# Barriers to adaptation of environmental sustainability in SMEs: A qualitative study

**Nazneen Durrani[1], Abdul Raziq[2], Tarique Mahmood [ID][3,4], Mustafa Rehman Khan [ID][3,5]\***

**1** Faculty of Management Science, Sardar Bahadur Khan Womens' University, Quetta, Pakistan,
**2** Department of Management Sciences, Balochistan University of Information Technology, Engineering and Management Sciences, Quetta, Pakistan, **3** Graduate Business School, UCSI University, Kuala Lumpur, Malaysia, **4** Management Studies Department, Bahria University Karachi, Karachi, Pakistan, **5** Faculty of Management Sciences, Shaheed Zulfikar Ali Bhutto Institute of Science and Technology, Karachi, Pakistan

\* mustafa.r.khan03@gmail.com

## Abstract

This study examines the antecedents of environmental sustainability in small and medium enterprises (SMEs) of a developing country and explores the specific internal and external factors for environmental sustainability. The study focused on SMEs in Balochistan, Pakistan, utilizing convenience and purposive sampling techniques to select a sample size of 30 SMEs. In-depth qualitative interviews were conducted using a semi-structured questionnaire. The results of the study revealed that lack of finance and education are major barriers to recognizing and addressing environmental sustainability issues, along with the lack of government support and regulations to ensure compliance with environmental safety laws, hence leading to low concern for sustainability practices among SMEs. Awareness and attitude of SME owners/managers, along with customer demand and government policies, influence the adoption of environmental sustainability practices. Overcoming financial constraints and promoting cooperation among stakeholders are key to fostering sustainable practices in SMEs. This research makes an important contribution to the sustainable management literature by providing new and in-depth insights into the barriers that impede environmental sustainability in SMEs of developing countries.

## Introduction

Sustainability is now a critical aspect of development, with growing awareness among people in developing countries [1]. Climate change and environmental disasters are major threats to the planet and must be addressed through policies that connect the environment, economy, and society for sustainable growth [2]. To prevent the environment, organizations are striving to adopt sustainable practices and focusing on green practices, processes, and products [3]. This enables firms to bring positive impacts on organizational performance and the environment. However, ambiguity is still high on how firms can contribute to sustainability, especially in context of small and medium enterprises (SMEs).

SMEs have not received significant attention in the ongoing global discussion about sustainable development [4, 5]. It is evident that large organizations impact environment. It is also crucial to acknowledge the pivotal role of SMEs as well. SMEs impact on environment

**Data Availability Statement:** All relevant data are within the manuscript and its Supporting Information files.

**Funding:** The author(s) received no specific funding for this work.

**Competing interests:** The authors have declared that no competing interests exist.

become clearer when we consider their collective impacts. Globally, over 95% of businesses are SMEs [6]. Although each SME may have little impact, the cumulative effect of SMEs can be substantial due to their large number [4]. It is often mentioned that SMEs are responsible for major pollution [7].

Environmental sustainability in SMEs refers to the practice of reducing the environmental impact of business operations while ensuring performance [8]. This can involve reducing waste, conserving resources, using renewable energy sources, and minimizing carbon emissions [8, 9]. However, SMEs face challenges in implementing sustainable practices due to limited resources, lack of access to information, and high initial costs [10]. This challenge becomes more severe in resource-constrained developing countries.

Researchers have been supporting the adaptation of sustainability practices [3, 11]. However, literature has not adequately addressed the sustainability issues faced by SMEs in developing countries as it has with developed ones [12]. This situation has been widely acknowledged in the literature [4, 5]. Wang et al. [10] shed light on the importance of sustainability initiatives in developing countries and the unique barriers and motivational factors. Similarly, Jabbour et al., [4] emphasized the significance of sustainability in developing countries and the need to study strategies for achieving sustainable practices. Additionally, Purwandani & Michaud [13] recognized the necessity to delve into sustainability matters within SMEs as it impacts business performance and presents new market opportunities.

Prior studies showed that organizations should focus on improving economic, environmental, and social sustainability through the triple bottom line approach [3, 14]. Albeit, the dynamics of a resource-constrained country are different, thus requiring a different approach to achieve environmental sustainability. Therefore, there is need to explore the barriers and limitations of a developing country before replicating those strategies that work well in developed countries.

In this study we address the gap by exploring the internal and external factors that contribute to achieve the environmental sustainability in SMEs of Pakistan. The present study recognizes the existence of environmental alerts in Pakistan, particularly in Balochistan that comprise great environmental concern and weak governance [15–17]. Hence, this research significantly contributes to the discipline of sustainability. Further, the research framework of this study encompasses environmental orientation, responsible environmental management, and eco-friendly practices, which grounded on strategic competencies such as pollution prevention, product stewardship, and sustainable development of natural resource-based view (NRBV) theory [18, 19]. Therefore, this study provides comprehensive understanding of internal and external dimensions of SMEs that need to be nurtured to improve environmental sustainability. While extending the previous research Baah et al. [18], this research provides new insight for top management, policymakers, governmental and non-governmental organizations (NGOs) for achieving environmental performance in SMEs of developing countries.

## Literature review

### Factors of environmental sustainability in SMEs

Research suggests that customers may prioritize low prices over environmentally friendly practices [20]. However, some argue that consumers prefer to buy from sustainable companies even if it means paying a slightly higher price [21]. Though research is not conclusive over consumer buying eco-friendly products from SMEs or big firms, however, it shows consumer preference for sustainable products. Since SMEs are a crucial component of the global economy providing a significant portion of GDP in countries [10]. Hence, it can be inferred that their collective impact on the environment is also substantial, contributing to 70% of overall

pollution [7]. Environmental sustainability is now a global concern for SMEs and comes under public policy limelight. SMEs may address environmental concerns through waste recycling or pollution prevention [22–24]. Therefore, addressing environmental considerations, it is important to consider both materialistic and non-materialistic approaches, such as production processes or long-term sustainability goals. The business community should also be mindful of resource scarcity and strive for a balance between current utilization and future needs, prioritizing long-term thinking over short-term profits.

SMEs prioritize survival and cost reduction that may not align with environmental sustainability [25]. The relationship between environmental sustainability and economic outcomes is complex and influenced by multiple factors. However, engaging in environmentally efficient practices can lead to long-term benefits and increased revenue through recycling, reuse and reduce [26]. Hence, it raised question whether eco-efficiency could increase profits for SMEs [27]. Researchers suggest that adaptation of sustainable practices and ecological partnerships can lead to financial success for SMEs [28]. Moreover, Hossain et al. [29] conducted a systematic literature review of articles published during 2009–2020 and identified 87 drivers of environmental sustainability and categorized them under internal and external dimensions. Researchers suggested that SMEs should focus on internal and external factors for implementation of sustainable practices. Hence, researchers identified internal and external barriers suitable to the context of this study. Table 1 summarizes the barriers and opportunities for SMEs.

In reality, adopting sustainability requires resources, however, SMEs already facing several challenges [30]. On the other side, SMEs strive to gain flexibility and adaptability [31], unfortunately SMEs often lack the resources to implement sustainable practices [10]. Researchers argue that factors such as time constraints, financial limitations, inadequate training, information access and a lack of environmental management are hindering SMEs in practicing environmental sustainability [13, 32]. A limited number of staff and routine work leave no time for learning and implementing sustainability measures [32]. Hiring professionals to train existing employees can help, however, financial constraints restrict SMEs to invest in knowledge and

**Table 1. Barriers and opportunities for SMEs.**

| Barriers For SMEs | Opportunities for SMEs |
|---|---|
| • Insufficient knowledge among owners/managers or employees.<br>• Limited workforce size.<br>• Constrained financial resources.<br>• Inadequate organizational capacity.<br>• Restricted and short-term business operations integrated into daily activities.<br>• Pessimistic belief that engaging in environmental sustainability practices might have a prolonged payback period.<br>• Non-compliance or lack of adherence to government legislation.<br>• Concerns about potential temporary cost increases for products, with owners fearing customer resistance to higher prices.<br>• Different levels of awareness among owners/managers.<br>• Diverse attitudes among owners/managers.<br>• Reluctance of owners/managers to alter their perspectives on environmental sustainability practices.<br>• Lack of awareness among stakeholders regarding the advantages of environmental sustainability practices compared to potential price increases.<br>• Limited knowledge about technological advancements and their potential benefits for overall SME performance. | • Implementation of laws and regulations mandating SMEs to adopt environmental sustainability practices.<br>• Flexibility in altering the existing setup.<br>• Proximity and strong ties with suppliers.<br>• Commitment of owner management towards incorporating environmental sustainability practices.<br>• Introduction of green innovative products.<br>• Utilization of part-time employees to align with new practices.<br>• Enhanced awareness about the benefits of environmental sustainability practices.<br>• Influence from community expectations regarding environmental sustainability.<br>• Increasing interest from investors in supporting and promoting environmental sustainability practices. |

ability enhancement. Another aspect is employee or manager reluctance towards green innovation due to tight schedule, lack of experience or latest knowledge [33]. Environmental practices require pro-environmental behavior of employees [34]. To address this problem, companies should establish a network and enhance their relationship with external partners [35], such as universities, to provide cost-effective training for their staff. Establishing short-term objectives may facilitate the achievement of long-term goals, allowing SMEs to foster internal sustainability.

The limited financial capital of SMEs creates a significant constraint in SMEs ability to invest in environmental sustainability practices. The impact of these investments on profits remains a lingering question. Despite the potential benefits, SMEs may struggle to sustain the financial burden of implementing environmental sustainability measures [36]. Consequently, SMEs can apply for grants and loans from government agencies and financial institutions to alleviate financial constraints, which, in return, help them to bring sustainability in process and increase revenue in long term. However, lack of financial support from government and financial institutions and formal lengthy procedures creates hindrances in adopting sustainable practices [13, 37]. Despite the awareness and willingness of owners and managers to embrace environmental sustainability, financial constraints prevent them from environmental initiatives or involving third parties [38].

Moreover, the decision to adopt environmental sustainability practices in SMEs lies with the owners and managers, who act as the decision-makers. Their perception of the benefits associated with pro-environmental behavior is a key factor in determining whether they choose to invest in such practices [34]. A positive attitude towards environmental sustainability is essential for owners and managers to initiate such practices in their firms [38, 39]. Manager's age determines the adaptation of sustainable practices and concerns for environmental performance [40]. Education and green entrepreneurial orientation also play a significant role in adopting sustainable practices in SMEs [35]. SMEs are mostly governed by their owners and family members hence they align their resources with their priority, and survival is the topmost priority. Thus, sustainability may be considered on priority in SMEs when owners, family member or managers have environmental instincts and can foresee that bringing sustainability is new way of survival.

On the other hand, the interests of stakeholders play a significant role in firm's approach towards environmental sustainability at different levels within the business [41]. Stakeholders often prioritize financial sustainability over social and environmental sustainability, as long as they receive some sort of return [42]. Large organizations can establish long-term partnerships with stakeholders and the resources to influence through marketing strategies. Albeit SMEs find it difficult to maintain the relationships in long-term and often have shorter-term relationships with stakeholders, thus making it difficult for SMEs to change their perceptions [25]. SMEs are supposed to have a responsibility to the community, customers, and employees they serve [37, 38, 43]. Jamali et al. [44] argue that SMEs operate on a personal level and are closer to their stakeholders due to work-family issues, trust, and employee retention, among other factors. Moreover, when owners see market forces and customers direction towards eco-friendly processes and products, they may shift their focus on sustainability. Researchers claim that owners may not consider sustainability issues until they face a problem and are not motivated to do so as long as they are reaping profits [45].

## Environmental sustainability practices and firm performance

The incorporation of sustainable practices in SMEs is influenced by various factors [10, 13]. These factors include the motivation of entrepreneurs to adopt certain practices based on their perception of potential positive financial outcomes or their concerns about the consequences

of not embracing eco-friendly approaches [46]. The actions taken towards sustainability can either positively or negatively impact the financial performance of SMEs. Some researchers suggested that implementing sustainability practices is considered an additional burden by SMEs [47], while others support that there is positive relationship between firm performance and environmental protection efforts [8, 36].

Environmental Management System (EMS) has proven to be highly beneficial for the sustainable development of SMEs. ISO 14001 offers certified guidelines to assist SMEs in reducing their environmental impact and achieving long term sustainability [48]. By implementing EMS, SMEs can lower energy consumption costs while also creating positive image in the eyes of customers, regulators, and the public. Furthermore, EMS enables SMEs to utilize latest green technology, which not only reduces costs and improves productivity but also minimizes their negative impact on the environment. This innovative approach to resource utilization, cost reduction and environmental preservation ultimately ensures the sustainability of SMEs [24, 49]. Additionally, when SMEs collaborate with businesses within their network and receive support from organizations, they become more environmentally innovative and effectively reduce costs [22]. This ability to leverage network advantages is crucial for SMEs. Subsequently, adoption of environmental practices allows SMEs to position themselves as excellent performers [50].

Researchers suggested that financial limitations were inversely related to the adoption of practices. However, it also revealed a correlation between access to capital and technological expertise with the implementation of sustainability measures regardless of company size [50]. Various approaches can be utilized to assess the impact of environmental sustainability practices on firm performance. These include enhancing stability, reducing emissions, improving eco-efficiency systems, conserving materials, energy and implementing ISO 14001 certification [9, 8, 24, 48]. Performance measures may involve creating environmentally friendly production processes and adopting innovative techniques. By adopting environmental sustainability practices firms can enhance their systems to minimize water usage, air pollution and toxic waste emissions [51]. Therefore, managers of SMEs who implement sustainability measures often believe in adopting innovative technologies [52].

The positive impact of effectively implemented EMS, on performance has been evident [48, 53]. Therefore, implementation of EMS leads to improved environmental performance, including reductions in water consumption and waste emissions. Previous studies found that the adoption of EMS practices provides benefits to SMEs [53]. Improvements in environmental performance and cost reduction go hand in hand. Eco-design plays a role in creating products with better recycling potential and improved performance. Furthermore, these eco-friendly designs are not only less harmful but also cost-effective to manufacture, leading to reduced expenses and improved firm performance [54]. When organizations strive to reduce pollution, it ultimately results in resource utilization and increased productivity. Sustainable practices like recycling and collaborating with suppliers contribute to cost reduction and enhanced efficiency. By utilizing metal scraps, waste materials, and recycled oil within SMEs operations along, with water recycling and purification measures, all contribute to cost reduction while simultaneously boosting profits.

Furthermore, implementing environmental protection initiatives can influence firm image [54]. Firms' performance heavily relies on their knowledge-based resources and the ability to effectively utilize these resources to gain advantages [52]. However, researchers argue that being an eco-friendly organization does not necessarily guarantee strong competitiveness in the market [50].

## Conceptual framework

This study proposes a research model to investigate sustainability in SMEs. Previous studies have examined how internal and external factors affect knowledge and attitudes toward environmentally conscious management practices [55]. This study included external and internal factors as independent variables, which encompasses suppliers, customers, and laws. These characteristics motivate and challenge management to adopt environmental sustainability initiatives. Stakeholder impact on SMEs is further influenced by environmental awareness and attitudes, which address top management understanding of the cost and benefits of being environmentally sustainable. These variables influence environmental sustainability practices, serving as dependent variable. Hence, external and internal factors influence the organization's attitude and understanding of environmental sustainability, which defines its policies, strategies, and systems for sustainable development. This process is influenced by owner-manager qualities, time, financial resources, and cost-benefit information about environmental sustainability. Without these criteria, SMEs may struggle to implement environmental sustainability initiatives. The effects of these factors need to be examined in the context of developing countries. Fig 1 depicts the research framework.

## Methodology

The target population for this study was SMEs, defined as firms with fewer than 250 employees [56], located in Balochistan, Pakistan. The industrial states included in the study were Quetta industrial and trading state, Hub industrial and trading state, Uthal industrial state, and Marble City. Due to the lack of proper records or censuses of SMEs in Balochistan, the researchers used convenience and purposive sampling techniques for data collection [57].

The data collection method used was a semi-structured interview based on an interviewer-administered questionnaire. This method was chosen because it was believed to provide in-depth understanding of subject matter [58]. The interview questionnaire was adapted from a similar study by [55] and revised based on a pilot study and expert opinions to ensure validity and reliability. This research adhered to ethical guidelines, securing respondent consent prior to interviews. Additionally, the authors sought ethical approval from the Institutional Review Board (IRB). However, the IRB returned the application with the recommendation that no ethical review is necessary for this research study. The research process followed in this study is presented in Fig 2.

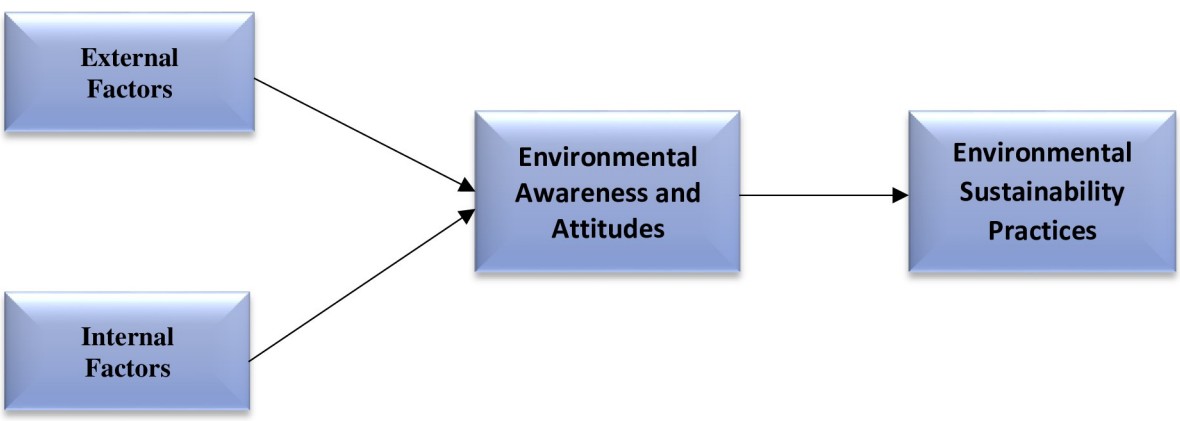

**Fig 1. Conceptual framework.**

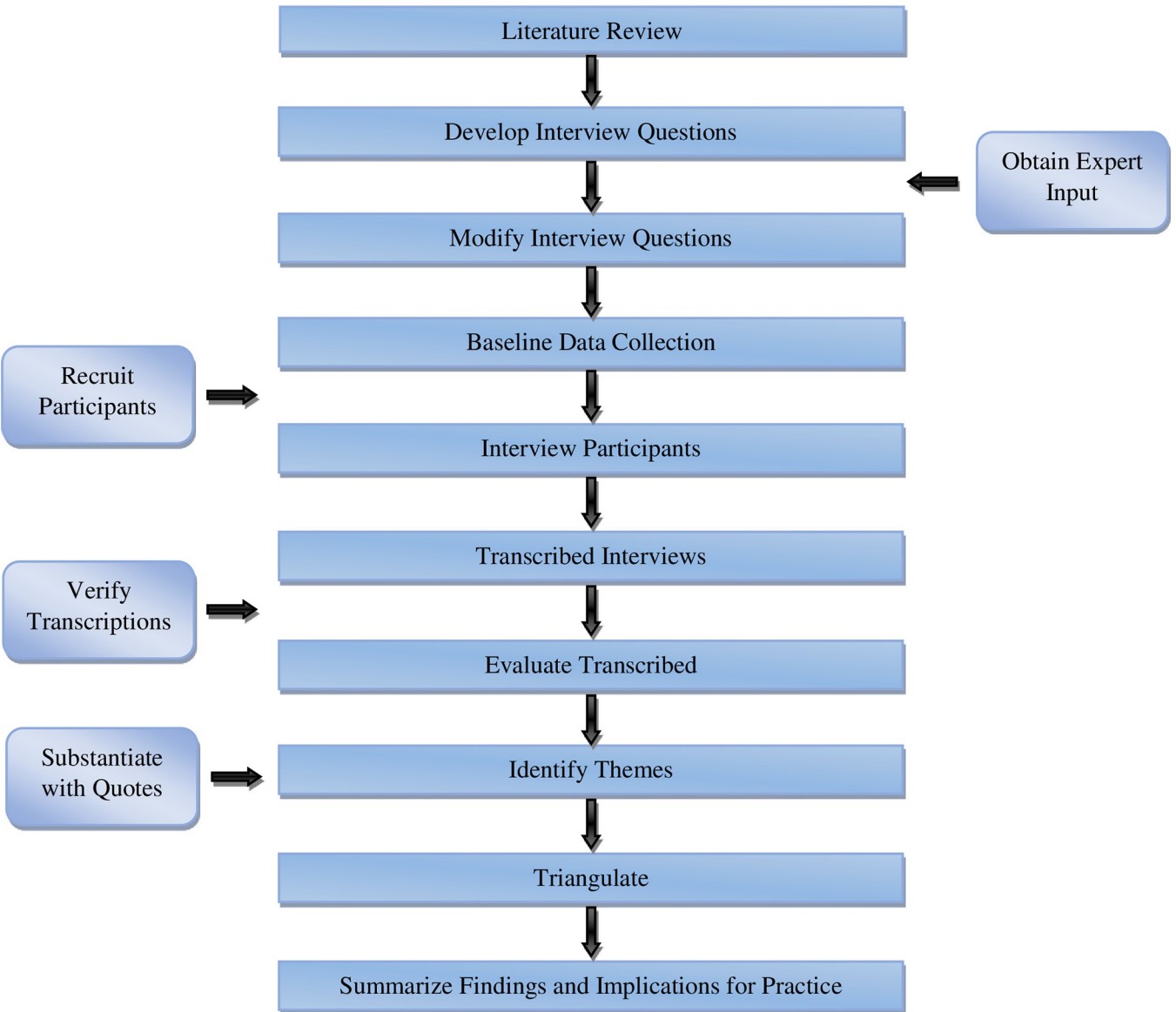

**Fig 2. Flow chart of research process.**

## Data collection

Data collection was carried out over a period of five months and involved a combination of convenience and purposive sampling techniques. The study began with a meeting with the head of Small and Medium Enterprises Development Authority (SMEDA) in Balochistan, Pakistan. The starting point for sampling was well-educated SME owners, and initial contact was made through phone calls. Appointments were set after presenting the interviewees with a letter of consent. The data collection process continued for several months in Quetta City, with the majority of manufacturing SMEs located in Hub City. As such, additional time was required for arrangements and contact. The head of the Chamber of Commerce in Hub and the Lasbela Industrial Estate Development Authority (LIEDA) were consulted for support and references to reach SME owners. Due to the conservative and underdeveloped environment in

Balochistan and the prevalence of terrorism, the respondents were only met when referred by a reliable reference.

During the interviews, a survey was conducted to determine the respondent's capability to understand, judge, and answer research questions. The survey questions were designed to obtain responses related to the adoption of sustainable practices in SMEs, and the interviews were conducted in person, demographic information of respondent is shown in Table 2. The interview method proved beneficial in providing diverse views and answers related to different businesses.

This study employed non-probability sampling technique; however, efforts were made to ensure that the sample was diverse and representative of the population of SMEs in the region. Moreover, due to the sensitive nature of the topic and the underdeveloped environment in Balochistan, the sample size was limited to 30 SMEs. However, prior studies recommended a sample size of 12–20 for qualitative research [59, 60]. Hence, the sample size of this study is deemed acceptable [61]. Despite these limitations, the present study provides valuable insights into the barriers to sustainable practice adoption in SMEs in Balochistan, Pakistan. Table 3 summarizes the respondent's answers to interview questions as "yes" or "no" to enable further discussion.

**Table 2. Demographic information.**

| | Characteristics | Frequency | Percentage |
|---|---|---|---|
| **Gender** | | | |
| | Male | 29 | 96.67 |
| | Female | 1 | 3.33 |
| **Sector** | | | |
| | Restaurant | 2 | 6.67 |
| | Hotel | 2 | 6.67 |
| | Furniture | 3 | 10.00 |
| | Agriculture | 2 | 6.67 |
| | Marble production | 5 | 16.67 |
| | Beauty parlor | 2 | 6.67 |
| | Motor Garage | 2 | 6.67 |
| | UPS manufacturers | 2 | 6.67 |
| | Salt company | 1 | 3.33 |
| | Steel plating | 1 | 3.33 |
| | Lubricants | 1 | 3.33 |
| | Wheel manufacturers | 1 | 3.33 |
| | PVC pipe manufacturers | 1 | 3.33 |
| | Service stations | 1 | 3.33 |
| | Dry item suppliers | 1 | 3.33 |
| | Shopping marts | 1 | 3.33 |
| | Gems and Jewelry suppliers | 1 | 3.33 |
| | Mosaic | 1 | 3.33 |
| **Location** | | | |
| | Quetta | 18 | 60.00 |
| | Hub | 10 | 33.33 |
| | Winder | 1 | 3.33 |
| | Khuzdar | 1 | 3.33 |
| **Position** | | | |
| | Owner | 15 | 50.00 |
| | Senior Manager | 15 | 50.00 |

**Table 3. Summary of responses.**

| Interview Questions | | Response | |
|---|---|---|---|
| | | Yes | No |
| **Awareness of environmental sustainability** | | | |
| 1. | "Is the term ecopreneur familiar to you?" | 2 | 28 |
| 2. | "We are aware of the solid and liquid waste management problems in the city." | 28 | 2 |
| 3. | "We are aware of the problems about city's sources of drinking water." | 28 | 2 |
| 4. | "We are aware of the problems about city's sources of electricity." | 28 | 2 |
| 5. | "We develop product or services with corresponding natural environmental impact in mind." | 4 | 26 |
| **External factors and environmental sustainability** | | | |
| 1. | "Firms communicate with customers/buyers about environmental issues." | 3 | 27 |
| 2. | "Firms deal with suppliers or distributors with environmentally friendly business practices." | 3 | 27 |
| 3. | "Role of government Legislations in infusing environmental sustainability practices in SMEs." | 4 | 26 |
| **Internal factors and environmental sustainability practices.** | | | |
| 1. | "Availability of adequate financial resources and availability of positive cash flows." | 11 | 19 |
| 2. | "Access to credit and loans from banks and other financial institutions." | 14 | 16 |
| 3. | "Can easily raise funds to support plans for expansion of production capacity?" | 11 | 19 |
| 4. | "Adequate financial resources to support further training and development of employees." | 4 | 26 |
| 5. | "Adequate financial resources to develop or buy new production machinery." | 10 | 20 |
| **Owners/managers attitude towards environmental sustainability practices** | | | |
| 1. | "We (i.e. managers and employees) have adequate knowledge about environmental sustainability and global warming, climate change etc?" | 6 | 24 |
| 2. | "Environment-friendly manufacturing practices are good for any business?" | 6 | 24 |
| 3. | "Which media has played its role in increasing awareness about environment issues?" | 30 | 0 |
| 4. | "Does religion teach the lesson of ethics?" | 5 | 25 |
| 5. | "Corporate social responsibility generates many profits?" | 10 | 20 |
| 6. | "Environmental sustainability should be present in in all organizations?" | 20 | 10 |
| 7. | "We know that business have important role to play in environmental protection." | 2 | 28 |
| 8. | "Is profit making better than care of environment?" | 27 | 3 |
| 9. | "Recycling of production waste is a normal practice in business?" | 15 | 15 |
| 10. | "The business gains more customers as a result of being environment-friendly business?" | 3 | 27 |
| **Environmental sustainability practices** | | | |
| 1 | "Company's policy regarding the environmental friendliness of the usage of resources in the business processes, such as water, gas, electricity and other natural resources." | 3 | 27 |
| 2. | "Policy regarding conservation of resources and reuse of waste materials?" | 26 | 4 |
| 3. | "Training of employees includes environmental awareness?" | 6 | 24 |
| 4. | "Firms practice in voluntary environmental programs?" | 6 | 24 |

## Results

The collected data from the semi-structured interviews was analyzed using coding techniques and condensed into main themes for further examination [62], presented in Table 4. The researcher repeatedly read through the data and utilized word clouds to identify sub-themes, consolidate common themes, and identify main themes, Fig 3 depicts the word clouds. A final review was conducted to identify any important or conflicting information related to the established themes.

### Theme one: Awareness of environmental sustainability

The initial stage of the research process involved an overview of environmental sustainability concerns and an assessment of the level of awareness among owners and senior managers of

**Table 4. Emerged themes.**

| Themes | Sub-themes |
|---|---|
| General Overview of SMEs' Environmental Awareness | Level of Education |
| | Sound knowledge |
| | Lack of education |
| | International Business Impact |
| Perception of Environmentally Friendly Practices | Long-Term Investments |
| | Conservation |
| External Factors Impacting Sustainability Practices | Influence of international customers on environmental policies |
| | Challenges with environmentally friendly practices |
| | Government involvement in promoting sustainability |
| | Adoption of sustainable practices |
| Internal Factors Impacting Sustainability Practices | Scale of Business |
| | Varied business performance across sectors |
| | Challenges in accessing credit and loans |
| | Training and development |
| | Lack of financial resources for environmentally safe machinery |
| Impact of Awareness and Attitudes on Sustainability Practices | Influence of education on understanding sustainability |
| | Different views on the relationship between corporate social responsibility and profit |
| | Willingness to adopt environmentally friendly practices for long-term sustainability |
| | Religious and ethical perspectives on environmental responsibility |
| Environmental Practices (Policies, Strategies, Systems) | Presence and nature of environmental policies |
| | Resource conservation and waste material reuse efforts |
| | Inclusion of environmental perspective in employee training |
| | Participation in voluntary environmental programs |
| | Challenges faced in implementing sustainability practices |

SMEs. The term sustainability was frequently interchanged with environmental care practices, but the concept of ecopreneurship was relatively unfamiliar to most of the interviewees. However, respondents with higher levels of education generally had a better understanding of environmental protection within their production processes. Meanwhile, those who were less educated did not possess any knowledge in this regard and did not adopt any environmental precautions in their business operations.

When asked about their understanding of the sources and problems of drinking water and electricity in their city, all respondents agreed that these issues were prevalent and that load shedding of electricity had significantly impacted their businesses. Some entrepreneurs were able to survive by relying on alternatives such as solar panels, but others had suffered significant losses. The scarcity of water was another challenge faced by most SMEs, both for drinking and business purposes. Despite the difficulties, most respondents felt that it was not their role as entrepreneurs to preserve water and electricity. Respondents highlighted that the cost of running business was already high, and the additional expense of water testing and purification only added to their burden.

One hotel owner said that *"running a hotel is a big responsibility and it increases our cost when we buy water from private vendors, if Quetta did not face the water scarcity problems, then we would also have benefited from it."*

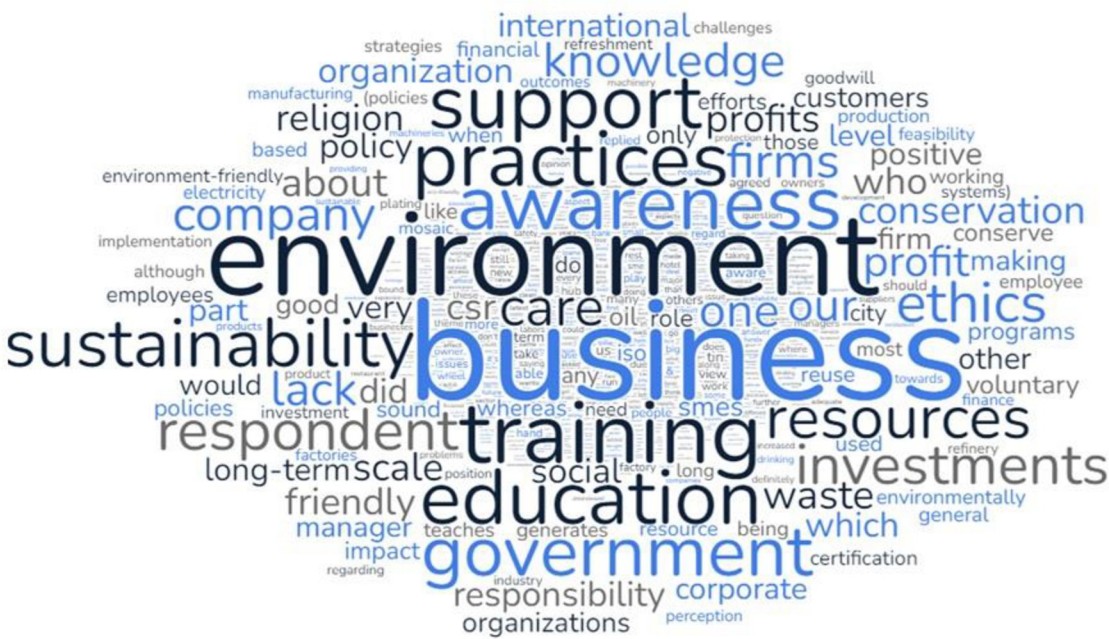

**Fig 3. Word clouds of responses.**

Another owner from a marble factory in Hub City said that *"although the water came from Hub Dam and its dependent on the level of rain. Water was there but it was not clean they had to arrange their own purification plants and filters, so that was also an expense."*

When discussing the firms' practices in developing products or services with consideration for their environmental impact, it was clear that the environment was not a significant concern for most SMEs. Very few firms had policies in place to minimize negative environmental impact, and these were mostly ISO-certified businesses that dealt with international clients. For the rest of the firms, their focus was on making enough profit to remain competitive in the market rather than considering the environmental impact of their operations.

Manager of steel plant said, *"It seems like we should adopt eco-friendly practices not just sake of environment but also for the benefit of our society as well as for profit."*

While the majority of SME owners tended to recognize the effects of businesses on the environment and the importance of being environmentally conscious, they also expressed their need for government support in this regard. The concept of sustainability was often misunderstood, with many respondents confusing it with environmental issues. However, when the term environmental sustainability was explained to them in detail, they were able to grasp its significance and the importance of considering this issue in their business operations.

### Theme two: External factors, environmental awareness, and attitude

The second theme of the study aimed to identify the impact of external factors on environmental sustainability practices. The firms were divided into two categories: those dealing with local customers and those dealing with international customers. Firms dealing with only local customers reported that cost was the primary concern of their customers, while firms dealing with international customers indicated that they were bound to follow certain policies, including environmental policies, to meet international business standards. The companies that were ISO-certified reported that it was mandatory for them to be environmentally sustainable in order to attract international customers.

One respondent said that *"its mandatory for them to be environmentally sustainable to capture customers at international level".*

SMEs faced a different set of challenges when dealing with environmentally friendly suppliers and distributors. If the SMEs had multiple choices of suppliers to select from or were bound by their customers to work with environmentally friendly businesses, they opted for environmentally friendly suppliers. On the other hand, SMEs in Balochistan province mainly dealt with local suppliers and distributors who lacked education and awareness about environmental sustainability issues. Most of the marble suppliers in this region belonged to poor communities and were not supported by any government agencies, making them bound to market trends in terms of supply.

The firms that had sufficient knowledge and were aware of the importance of environmental sustainability had adopted methods to be environmentally sustainable, but not to their full potential due to the lack of support from government organizations. They expressed a desire to work towards environmental protection but faced challenges due to the lack of awareness and adoption of environmentally sustainable practices in the supply chain. The focus on cost reduction rather than environmental protection also presented an obstacle. According to the results of the study, the government in Quetta seems to have little to no involvement in promoting environmental sustainability among SMEs. The respondents reported a lack of awareness of any governmental institutions or initiatives aimed at promoting environmentally safe practices.

One respondent said that *"now a day's metropolitan of Balochistan had signed a contract with the Turkish company to manage recycling of waste."*

According to this new contract, counselors in every area would need to sign a contract and agree to pay a fund in the earliest days to pick up and recycle the waste for few early years, and after some time, when the company gets established, the Turkish company instead of taking money would pay money for managing waste products.

Respondent from oil manufacturing firm said that *"LIEDA is playing its role of waste management and EPA (Environmental protection agency) occasionally shows its concern in Hub".*

## Theme three: Internal factors, environmental awareness, and attitude

The business performance of various sectors was assessed, and the results were varied. While some businesses, such as restaurants, lubricants, rice dealers, and furniture, were reported to be operating successfully and were financially stable, many others were struggling to survive and expressed concern about their current situation.

A respondent from a tin making firm said that *"my business had suffered losses in the past few years but had reached a breakeven point the previous year. However, the business remained at risk as tin was being rapidly replaced by plastic."*

The manager of a mosaic making firm reported a sluggish financial situation, as the cessation of funding from a donor had resulted in increased expenses while their revenue remained stagnant. The manager attributed this to the low level of awareness and advertising of mosaic products, which primarily cater to high-end society and hotels.

Most of the respondents from the marble factories reported that the industry was facing stiff competition and was plagued by high waste due to a lack of proper machinery. This led to many marble factories closing due to losses, with only a few who had access to modern machinery and cutting-edge technology being able to remain profitable. The remaining businesses were fighting for survival.

SMEs are facing challenges in accessing credit from banks and other financial institutions due to difficult and lengthy bank procedures, high interest rates, and the risk of losing their businesses to repay loans. Some of the SMEs interviewed had positive views about accessing

credit, saying that if they were registered and had an NIT number, they would easily get loans. However, even these SMEs were hesitant due to the burdensome procedures involved. A few respondents mentioned that if they had personal contacts with bank officials, they would not face any problems accessing credit. On the other hand, some of the SMEs had negative views and highlighted that they were unable to meet their monetary needs, and they faced difficulties in accessing credit and loans.

Respondents from the marble factory said that *"banks nearly took over our business as factory was kept as collateral"*.

The CEO of a mosaic making firm said that *"SMEs in the mosaic making industry face many difficulties in accessing financial institutions."*

When asked about the possibility of raising funds for expanding production capacity, the SMEs were divided. Some SMEs in a sound financial position were capable of raising funds from their businesses, family, friends, and relatives, while others were struggling to survive and had no intention of expanding.

The owner of marble company said that *"we thought to easily raise funds from the business community and friends, but the current conditions in the Marble City and the downward trend of business made it difficult to attract any further investments"*.

On the other hand, a manager from a salt making firm said that *"the company had a mission to reinvest, and it could easily raise funds for expansion from equity or risk."*

Training and development of employees were also a concern for the SMEs. The firms in a sound financial position were willing and able to support their employees, while those struggling to survive were reluctant to do so.

The owner of a rice distribution firm said that *"we sent their employees for further education and paid training to keep them competitive and aware of modern business techniques"*.

The wheel making and salt refinery firms had training as part of their regular policies, and employees were given training for promotion and continuous learning. However, only a few of the thirty firms interviewed considered their employees valuable assets and invested in their training and development. The availability of financial resources to develop or buy new, environmentally safe production machinery was another concern.

The respondent from a tin plating firm said that *"the company wants to take a bank loan for the long-term profitability and better production machinery."*

The manager of a salt refinery firm said that *"expertise in machinery was not available in Pakistan, so we are working on upgrading machinery from foreign sources for better production and environmental safety"*.

Some of the oil refining firms made small investments in new equipment for innovation. However, the majority of the marble factories were suffering from a lack of investment in new machinery and did not have government support in this regard. Only a few marble factories that were able to invest in new machinery were earning more profits and were environmentally safe.

One owner from a marble factory said that *"we are not interested in investing in new machinery as the business is not future-promising. Although we are in the position to afford it."*

This is contradictory situation where, at one side, SMEs claim that due to lack of financial resources, they are not able to invest in latest machinery, on the other side, some firms can afford but only invest to see the profit prospect of the business, not the environmental sustainability.

## Theme 4: Environmental awareness, attitudes, and environmental sustainability

The culture of an organization is shaped by its employees, with senior managers and owners setting the standards and policies. When asked about their knowledge of environmental

sustainability and climate change, the level of education of the respondents was found to be a facilitator or a hindrance. Those with a higher level of education had a better understanding of the topic, while others were less informed. The manager of a mosaic-making firm, who was MPhil scholar, was proud of his company's adoption of environmentally sustainable practices.

He said that *"the company uses leftover marble residuals in the production of mosaic in partnership with an international donor company to turn waste into profit"*.

The majority of respondents agreed that being environmentally friendly is good for business. However, a minority who did not agree were conservative in their approach and lacked education and awareness of the benefits of eco-friendly business. The respondent from the mosaic manufacturing SME saw conservation and precautions as long-term investments, while another from a marble factory expressed concern for the environment and the impact it would have on their business if they did not adopt environmentally friendly practices. SMEs that did business internationally, such as marble, mosaic, and salt, stated that being environmentally friendly was part of doing business. These firms reported that international companies refuse to do business with those who do not have an environmental care aspect in their business. The media, including television and the internet, was acknowledged by the respondents for increasing awareness of environmental issues.

Nearly all respondents agreed that religion teaches the importance of ethics and environmental sustainability. For example, one respondent from the marble factory said that their religion emphasizes cleanliness and that being socially and environmentally responsible is part of their belief.

Manager from the oil industry said that *"In Islam the first and foremost thing is care of humans and for that ethics bound us to take care of our environment. Religion Islam gives us the teaching that one human should not hurt the other, so if we act upon it then according to this care of others, care of society and care of environment is definitely part of our ethics"*.

Most of the firms interviewed saw corporate social responsibility as a tool for creating good will among the public. The respondents from the restaurant industry said that; *"According to me the answer is definitely corporate responsibility generates more profits because in our business it all depends on the care of customers, providing them with fresh & healthy food in a healthy & clean environment is the core of our business & that's possible only if we are careful about the corporate responsibility"*.

Established SMEs, such as the tin plating firm, made efforts to fulfill their corporate social responsibilities by planting a water purification plant in a village suffering from polluted water and by running schools. The oil refining firm saw corporate social responsibility as a long-term investment, as it prevents the spread of diseases like hepatitis among their workers and their families. The Marble City respondents expressed their efforts to fulfill their responsibilities but faced hurdles from uneducated workers and non-cooperative labor. Four respondents did not agree that corporate social responsibility and profit margins go together. The manager from the salt industry saw it as a matter of economics rather than social responsibility and stated that their company provides medical facilities and schools for their workers. They believed that thinking about profit and social responsibility at the same time leads to a conflict of interest. All of the participants had a positive opinion, saying that environment is preferable to us, as this is where we live and want it to be secured. They support environmental sustainability being present in all firms, although it may reduce profit.

One respondent said, "*As we deal with rice supply, we used to pack our rice in plastic bags but after realizing that this is harmful for the environment we changed to paper bags for packing. Although it increased our cost but for long-term sustainability it seemed more preferable.*"

The respondent working at the Parlor had an opinion regarding the presence of environmental sustainability in organizations by saying that "*In our business environmental*

*sustainability does not decrease profit rather, it gives us competitive advantage, when our customers observe us using environmentally friendly products, they come more often and spread good words, resulting in increased customers and more profit.*"

One of the business owners stated that business was done for profit, but it was also necessary to care for the environment as it was the home where the business was located. The environment and the earth were the sources of all the resources used in production and it was impossible to move forward without taking care of the environment. Half of the respondents agreed to recycle their production waste when questioned about it, while the other half either had no waste or were unaware of recycling practices. The manager of a tin plating company reported that the water used in their process was recycled and the tin scraps and hardboard leftovers were sold to packing companies for box making. The gems and jewelry company reused their residual materials in fine jewelry and synthetic items. The respondent from a salt company reused salt waste by evaporating water using the sun's heat. An oil refinery invited experts from Switzerland to help manage their waste and recycling efforts and had reclamation plants to clean dirty oil for reuse. However, the used oil did not receive a proper price. Finally, the respondents who were already working at an international level stated that being environmentally friendly was necessary to do business at an international level. They said that being ISO certified was important to gain international customers as it required firms to operate in an environmentally safe manner. The same was true for restaurant businesses, as they were required to follow policies and criteria to do business at an international level.

## Theme five: Environmental practices (policies, strategies, systems)

Most firms interviewed, except for three SMEs, did not have a written policy on environmental sustainability practices. The firms with environmental policies only did so to meet ISO certification requirements. The manager of the wheel manufacturing firm reported having an ISO certification for two years, which includes environmental policy. The oil refinery company also had an environmental policy, which they upgraded periodically.

When asked about resource conservation and waste material reuse, the owners and managers gave varying answers, but the message was consistent that they strive to conserve resources but face opposition from their workers. The manager of the tin plating company said they are trying to replace inefficient heaters with more efficient ones, but their workers are uncooperative. The wheel manufacturing firm has made efforts to conserve water and electricity by switching to energy-saving bulbs. The oil refinery firm efficiently uses the boiler to maximize production. The respondent from Marble City said that despite efforts to conserve electricity and water, uneducated workers often ignore the advice, causing problems for the company. Overall, the respondents agreed that resource conservation leads to cost reduction.

Many SMEs provided general training to their employees on the usage of machinery and chemicals, however, only some of them included environmental perspective in their training. For example, the respondent from the tin plating company stated that their employees were trained to handle chemical waste and use machinery in an environmentally conscious manner, but most of them being uneducated, did not understand the significance. The manager of the oil refinery company mentioned that they did provide training on environmental aspects, as even a small oil leak could cause significant land pollution, which they aim to avoid.

The respondent from the mosaic firm added that their international organization required them to train their employees on various subjects, including the environment. Most of the respondents acknowledged that they tried to guide their employees on environmental protection, but the lack of education was a major hindrance to understanding the concept. However, one of the marble factories, being a big organization, provided step-by-step training to all their

employees on various aspects. The owner of the furniture company stated that their employees were trained based on their level of knowledge.

When asked about voluntary environmental programs, the participation of SMEs varied depending on their financial position and level of environmental awareness. For instance, the tin plating company had a water purification plant to conserve water, while the salt-making company focused on providing health care facilities and education rather than environmental programs. The oil refinery company mentioned their hepatitis vaccine campaign as a voluntary program to save the environment from contagious diseases. One of the respondents from Marble City said that his organization was working very efficiently in voluntary environmental programs like tree plantations, etc., but water consumption and costs of purchasing water increased, and he was not able to continue it. The mosaic-making firm stated that their voluntary program involved training females on how to reuse and conserve resources, maximize their income and reduce waste. However, one of the respondents from Marble City mentioned that although their organization was working efficiently on voluntary environmental programs such as tree plantations, the increased water consumption and purchasing costs made it difficult for them to continue these initiatives.

## Discussion

The study aimed to understand the level of awareness among owners and senior managers of SMEs regarding environmental sustainability issues. The results are aligned with strategic abilities of NRBV theory, such as pollution prevention, product stewardship, and sustainable development. NRBV provides support for the theme of this study, which encompasses environmental-orientation, responsible environmental management, and eco-friendly operations.

The qualitative analysis revealed that majority of SMEs owners or managers were not familiar with the term "ecopreneurship." This was primarily due to their low level of education, knowledge, or lack of concern for environmental issues. Researcher argues that environmental practices rely on equal involvement from all employees and lack of knowledge hinders their implementation [20]. In Balochistan, the shortage of electricity, water, law enforcement and mismanagement are well known [16]. Every respondent interviewed acknowledged this problem and agreed on the need to conserve these resources. However, the awareness was limited to owners and managers, and they faced difficulties in making their workers understand and address sustainability.

Most SMEs, except for a few that were ISO certified, did not consider environmental impact in their business practices. Firms that obtained ISO certification only did so to expand their international business, as certification was mandatory for international business dealings. These findings are consistent with previous literature suggesting that unawareness and lack of laws are major reasons for environmental harm [63]. When law enforcement is weak, people tend to harm the environment. However, strict regulations and the fear of punishment may prompt SMEs to make changes that help them understand the costs and benefits of their actions. Hence, external pressure such as regulatory compliance and legal certainty, is a primary motivator for firms to adopt environmental practices [18, 32]. Thus, it is important for customers, governments, and supportive organizations to work together to educate and encourage SMEs to adopt environmentally sustainable practices.

Sustainable practices play a critical role for businesses in boosting their market share. Researchers suggested that sustainable practices offer benefits such as enhanced market position, increased customer base, and competitive edge [64]. In line with previous studies, this study identified that SMEs that are ISO-certified believe they must be environmentally sustainable to capture international customers [37, 38, 43, 44]. Further, external factors, such as

customers, can act as both motivators and obstacles for SMEs in terms of adopting environmentally sustainable practices. However, local customers who are price-sensitive may be hesitant to accept the potential price increases that come with environmentally sustainable practices. Meanwhile, SMEs in Balochistan province face further hurdles as their supply chains come from poor backgrounds, lacking financial resources and government support, forcing them to prioritize their livelihood over environmental sustainability [65].

The adoption of environmental sustainability practices by SMEs is a subject of debate due to their limited financial capital. It remains unclear whether such investments will result in increased profits or whether SMEs may face financial challenges when engaging in such practices [36]. On the other hand, applying for grants and loans from government agencies, energy service companies, or non-government organizations can help SMEs overcome financial constraints and pursue environmental sustainability [66]. However, some studies have suggested that the willingness to adopt environmental sustainability practices has nothing to do with the economics or size of the firm but rather depends on the awareness and attitude of the company [67]. In the case of the SMEs in Balochistan Province, financial constraints were the main reason for their reluctance to invest in environmentally sustainable practices. While they could have sought financial help from personal sources, the complicated bank process and the prevailing business and security conditions made them hesitant to do so.

The positive mindset and attitude of the owners are crucial in driving environmental sustainability initiatives within the firm [38, 39]. SMEs, being owned and operated by their founders, can align their resources with their priorities, including prioritizing long-term sustainability over short-term profits [37]. If they see benefits in such environmental practices, they may allocate finances towards them, but if they view it as a long-term investment, they may be reluctant to adopt even if they have the means to do so [68]. The owners and managers may focus on certain sustainability aspects while disregarding others, depending on their mentality [69]. However, some owners may only consider sustainability issues when they face problems and otherwise prioritize profits over sustainability [66, 67].

The limited knowledge of SMEs owners and managers may lead them to believe that pro-environmental measures are costly without considering the long-term benefits, such as the reduction of toxic waste and energy consumption through eco-friendly machinery and practices like using energy-saving light fixtures [70]. Researchers contended that further incentives are needed to convince SMEs owners to take bolder pro-environmental measures, even if they have already taken initial steps [71]. Moreover, the study highlights that religion plays a role in shaping attitudes toward environmental sustainability. Nearly all religions teach similar moral values, but the intensity and nature of belief may vary [72]. Researchers showed a correlation between religion and environmental sustainability [73], where owners in Balochistan agreed that religion teaches ethical values, but finances, lack of education, and knowledge limit the practical application of these values.

These findings are in collaboration with previous studies that support a positive correlation between environmental awareness and practices among SME owners/managers [22, 52, 55]. This study suggests that SMEs may struggle to implement environmental management practices due to a lack of awareness of their environmental impact. Moreover, environmental awareness is influenced by legislation, leading to formal compliance systems and changes in conservation behavior within SMEs [53]. The majority of SMEs owners and managers lack the education and research environment to adopt pro-environmental practices, and their attitudes are shaped by media exposure. Some established SMEs practice corporate social responsibility, while others view it as separate from profit-making or as a burden on their income.

## Managerial implications

The study provides significant implications for SMEs owners, managers, government, and environmental agencies. First, it is utmost important for both owners and workers to have access to education regarding environmental initiatives and their advantages. Closing the knowledge gap through workshops and training programs is essential, not only for aware SMEs but also to make them stick to eco-friendly protocols. Additionally, government agencies and environmental NGOs should advocate for increased government support and incentives as financial limitations pose a barrier. Second, SMEs should view ISO certification not as a requirement but also as a strategic tool that provides an advantage in accessing international markets. SMEs should engage with customers to raise awareness about the benefits of eco-friendly practices, which can help address customers' concerns about increased prices. Third, SMEs should position themselves as environmentally friendly organizations that open opportunities for financial support mechanisms such as environmental initiative grants and loans. Fourth, managers should develop CSR strategies that align with SMEs priorities, which create shared value and enhance reputation. Fifth, managers should collaborate with other environmental agencies and establish networks with NGOs and governmental agencies to amplify the impact of sustainability initiatives. Managers should also explore the opportunities for adaptation of climate friendly technologies and processes. Sixth, government agencies should take initiatives to foster a mindset among SMEs owners by emphasizing how sustainability aligns with long term profitability. Finally, government agencies should develop policies to monitor and evaluate environmental initiatives and performance of SMEs, which ensures improvement and adaptation to environmental initiatives. These research implications contribute to long term success and sustainable development of SMEs.

## Conclusion

This study examines the impact of internal and external factors on the environmental sustainability of SMEs in Balochistan, Pakistan. The qualitative research method was used to gain a deeper understanding of the motivators and barriers towards environmental sustainability. This study indicated that adoption of environmental sustainability practices by SMEs is influenced by a combination of internal and external factors. Further, awareness and attitude of the company owners and managers play a critical role in implementation of environmentally sustainable practices. Internal factors, such as financial constraints and lack of awareness, were identified as the primary reasons for the limited adoption of environmentally sustainable practices among SMEs in Balochistan province. However, the study also found that applying for grants and loans from government agencies or non-government organizations can help SMEs overcome financial constraints and pursue environmental sustainability. Further, external factors, such as customers and government policies, play a crucial role in promoting environmental sustainability among SMEs. In the similar vein, these internal and external factors significantly influence eco-friendly attitude and awareness of SMEs owners or managers, which leads to overcome barriers such as lack of knowledge, polices, training, and foster positive attitude for adoption of environmental sustainability practices within SMEs.

## Supporting information

**S1 Data set.**
(XLSX)

## Author Contributions

**Conceptualization:** Nazneen Durrani.

**Formal analysis:** Nazneen Durrani.

**Funding acquisition:** Tarique Mahmood.

**Methodology:** Nazneen Durrani.

**Project administration:** Tarique Mahmood.

**Supervision:** Abdul Raziq.

**Writing – original draft:** Mustafa Rehman Khan.

**Writing – review & editing:** Tarique Mahmood, Mustafa Rehman Khan.

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
