## [Decision Letter · Decision Letter 0]

5 Dec 2023

PONE-D-23-22784Unlocking the Barriers: Exploring the Impediments to Environmental Sustainability in Developing Country SMEsPLOS ONE

Dear Dr. Khan,

Thank you for submitting your manuscript to PLOS ONE. After careful consideration, we feel that it has merit but does not fully meet PLOS ONE’s publication criteria as it currently stands. Therefore, we invite you to submit a revised version of the manuscript that addresses the points raised during the review process.

Please carefully check the reviewers' comments and make necessary improvements to the manuscript. The reviews provide valuable insights into areas that require attention. 

We look forward to receiving your revised manuscript.

Kind regards,

Agnieszka Konys, Ph.D.

Academic Editor

PLOS ONE

Journal Requirements:

2. You indicated that ethical approval was not necessary for your study. We understand that the framework for ethical oversight requirements for studies of this type may differ depending on the setting and we would appreciate some further clarification regarding your research. Could you please provide further details on why your study is exempt from the need for approval and confirmation from your institutional review board or research ethics committee (e.g., in the form of a letter or email correspondence) that ethics review was not necessary for this study? Please include a copy of the correspondence as an ""Other"" file.

4. Please ensure that you include a title page within your main document. You should list all authors and all affiliations as per our author instructions and clearly indicate the corresponding author.

Reviewers' comments:

Reviewer's Responses to Questions

**Comments to the Author**

1. Is the manuscript technically sound, and do the data support the conclusions?

Reviewer #1: Partly

Reviewer #2: Partly

Reviewer #3: Yes

2. Has the statistical analysis been performed appropriately and rigorously? 

Reviewer #1: No

Reviewer #2: Yes

Reviewer #3: N/A

3. Have the authors made all data underlying the findings in their manuscript fully available?

Reviewer #1: No

Reviewer #2: Yes

Reviewer #3: Yes

4. Is the manuscript presented in an intelligible fashion and written in standard English?

Reviewer #1: Yes

Reviewer #2: No

Reviewer #3: Yes

5. Review Comments to the Author

Reviewer #1: The paper is well-written and provides a clear and concise overview of the study. It is well-organized and easy to read, and it effectively highlights the key findings and contributions of the research. However the paper is written in a report format. The following revisions should be made:

1. Add a flowchart/diagram representing the process study made.

2. Supporting results in form of tabular data/graphs can be added.

3. In the findings section, consider providing more specific examples or details to support the key findings.

4. In the practical/implications section, consider discussing the specific implications of the research for SME owners/managers, customers, and government policymakers.

Reviewer #2: Dear authors,

Thank you for the opportunity to review this very interesting piece. While I believe your work is worth publishing, there are several issues that I would like to address prior to this.

First Extensive editing of the English language and style is required.

The title is confusing Please re-write.

Abstract

Please concise abstract of the study.

Introduction

- Set the stage for context, and history, and summarize what you’re going to do. Leave the reader with some impression as to the importance of the topic.

- There are many writing errors such as“ (Even though the research on environmental sustainability is growing, there is gap in how SMEs can achieve environmental sustainability especially in context of recourse constraint developing country.”) What is this? Where is in/of?

- Why there is a need to explore the barriers and limitations of a developing country? There should be a strong background please pay attention to this part.

Literature review

- You used too many headings. Please read some articles to get a paper writing idea.

- The literature review in the study should be like a logical story that tells the readers about the gap that you intend to fill. What does the existing literature tell us? What do we know and don't know based on literature?

Methods and Results

- In the data collection you do not need to add the aim of the study: (The present study aimed to investigate the barriers to sustainable practices adoption in small and medium-sized enterprises (SMEs) in Balochistan, Pakistan.) You should discuss this in the introduction.

- I appreciate the results description, but again in the results, there are too many headings. I suggest you read some papers to follow the structure.

Discussion and Conclusion

- The discussion must be rewritten considering the findings. The author should have a higher deepness in the discussion, making the linkage to the results of other papers and the theory.

- Overall manuscript has replication please avoid it when revising.

- Spell out the full term at its first mention, indicate its abbreviation in parenthesis, and use the abbreviation from then on. Such as Small and Medium-sized Enterprises (SMEs).

Reviewer #3: The topic is important and timely. The methodology is unique as well but need further improvement in all sections to meet the standard of quality publication.

1) Abstract is comprehensive.

2) “SMEs can play a significant role in promoting environmental sustainability,”..provide justification that why SMEs are deserving priority over large firms?

3) In the literature review researcher highlighted the internal and external factors nicely however I believe there are more external factors can be incorporated apart from stakeholder’s aspects such as community/societal influence, buyers pressure, market dynamics and so on. Internal barriers part also can be enhanced. I highly encourage author (s) to read this paper to improve these lacking.

-Hossain, M. I., Ong, T. S., Tabash, M. I., Siow, M. L., & Said, R. M. (2022). Systematic literature review and future research directions: drivers of environmental sustainability practices in small and medium-sized enterprises. International Journal of Sustainable Economy, 14(3), 269-293.

4) Why the scope chosen only Balochistan? Justify. And importance of the research on Baluchistan needs to incorporate in introduction section, why it’s so crucial and urgent.

5) “due to the sensitive nature of the topic and the underdeveloped environment in Balochistan, the sample size was limited to 30 SMEs.”…these are not well convincing reason …how about the saturation point, did that met and how you ensure that no need further data collection?

6) Provide definition of SMEs in your context in methodology section.

7) 4-star hotel is this a SME? Confirm. That’s why you need to mention the criteria of choosing SMEs.

8) Mention the question asked in the interview.

9) The quotation from informants needs to add (not all quotation and no need full only key statements) where suitable otherwise it shows only researchers opinion.

10) Better to have a graphical overview/ model of nodes/codes and emerged themes.

11) “Awareness and attitude of the company owners and managers play a critical role in determining the level of engagement in environmentally sustainable practices.” Suggest how to enhance the level of engagement.

12) Managerial implication need to be aligned with the findings of the study rather than just mention generic statements.

13) Conclusion section needs to be more streamed and concise.

14) Pls consult with professional proof-reader to improve the language quality.

6. PLOS authors have the option to publish the peer review history of their article (what does this mean?). If published, this will include your full peer review and any attached files.

Reviewer #1: No

Reviewer #2: No

Reviewer #3: No

---

## [Author Response · Author response to Decision Letter 0]

22 Jan 2024

RESPONSE SHEET

Manuscript ID: PONE-D-23-22784.R1

Dear Editor,

We would like to thank you for considering our manuscript submitted to PLOS ONE Journal for possible publication and providing us an opportunity to revise and resubmit our manuscript based on the reviewers’ comments. We appreciate and acknowledge the inputs offered by reviewers that have helped us to improve the presentation and quality of the manuscript.

In the report below we have described the actions we have taken in response to each query raised by reviewers. We hope that you will find the changes we have made to be sufficient for acceptance of the paper.

With regards,

On-behalf of all the co-authors

Action Taken Report

Dear Reviewers,

We would like to express our sincere gratitude to you for devoting your quality time spent on reviewing our manuscript and offering us detailed inputs which have been extremely helpful in shaping our manuscript. 

We hope you will appreciate our sincere efforts in revising the manuscript. Below, we present our responses and actions in tabulated form.

 Editor Comments Author's Response

 1 Please ensure that your manuscript meets PLOS ONE's style requirements, including those for file naming. The PLOS ONE style templates can be found at 

Thank you. We have followed the formatting guidelines of PLOS ONE. 

2 You indicated that ethical approval was not necessary for your study. We understand that the framework for ethical oversight requirements for studies of this type may differ depending on the setting and we would appreciate some further clarification regarding your research. Could you please provide further details on why your study is exempt from the need for approval and confirmation from your institutional review board or research ethics committee (e.g., in the form of a letter or email correspondence) that ethics review was not necessary for this study? Please include a copy of the correspondence as an ""Other"" file. 

Thank you for your comment. This study is quantitative, the authors conducted interviews with managers and owners of SMEs to explore their awareness and attitudes toward sustainability practices in business. Furthermore, the study did not involve physical testing or direct contact with respondents. Consequently, there is no possibility of harm—whether mental or physical—to the participants. This absence of risk enables the authors to conduct the study without implementing any precautionary protocols, distinguishing it from qualitative studies in medical research. 

Further, we have provided an official letter from the review board stating that no ethical approval was required for this research paper.

3 In your Data Availability statement, you have not specified where the minimal data set underlying the results described in your manuscript can be found. PLOS defines a study's minimal data set as the underlying data used to reach the conclusions drawn in the manuscript and any additional data required to replicate the reported study findings in their entirety. All PLOS journals require that the minimal data set be made fully available. For more information about our data policy, please see http://journals.plos.org/plosone/s/data-availability.

Thank you. We have uploaded data set underlying the results of this study.

4 Please ensure that you include a title page within your main document. You should list all authors and all affiliations as per our author instructions and clearly indicate the corresponding author. 

Thank you. We have uploaded the title page as per author's instructions.

5 Please include your full ethics statement in the ‘Methods’ section of your manuscript file. In your statement, please include the full name of the IRB or ethics committee who approved or waived your study, as well as whether or not you obtained informed written or verbal consent. If consent was waived for your study, please include this information in your statement as well. 

Thank you. We have included a statement indicating that ethical approval was waived by the Institutional Review Board, as shown below:

Page 6, Lines 35-38

This research adhered to ethical guidelines, securing respondent consent prior to interviews. Additionally, the authors sought ethical approval from the institutional review board (IRB). However, the IRB returned the application with the recommendation that no ethical review is necessary for this research study.

Reviewer 1 Comment Author's Response

1 The paper is well-written and provides a clear and concise overview of the study. It is well-organized and easy to read, and it effectively highlights the key findings and contributions of the research. However the paper is written in a report format. 

Thank you for your comment. We have followed your instructions to improve our article.

2 Add a flowchart/diagram representing the process study made. 

Thank you. We have included a figure representing the flow of the research process on page 7, shown in figure 2.

3 Supporting results in form of tabular data/graphs can be added. 

Thank you for your comment. We have incorporated the results in tabular form on pages 9 and 10, in Tables 2 and 3.

4 In the findings section, consider providing more specific examples or details to support the key findings. 

Thank you for your comment. We have provided specific examples in support of our findings.

5 In the practical/implications section, consider discussing the specific implications of the research for SME owners/managers, customers, and government policymakers. 

Thank you for your comments. We have improved the research implications on page 19, lines 19-41.

Page 19 Lines 19-41.

The study provides significant implications for SMEs owners, managers, government, and environmental agencies. First, it is utmost important for both owners and workers to have access to education regarding environmental initiatives and their advantages. Closing the knowledge gap through workshops and training programs is essential, not only for aware SMEs but also to make them stick to eco-friendly protocols. Additionally, government agencies and environmental NGOs should advocate for increased government support and incentives as financial limitations pose a barrier. Second, SMEs should view ISO certification not as a requirement but also as a strategic tool that provides an advantage in accessing international markets. SMEs should engage with customers to raise awareness about the benefits of eco-friendly practices, which can help address customers’ concerns about increased prices. Third, SMEs should position themselves as environmentally friendly organizations that open opportunities for financial support mechanisms such as environmental initiative grants and loans. Fourth, managers should develop CSR strategies that align with SMEs priorities, which create shared value and enhance reputation. Fifth, managers should collaborate with other environmental agencies and establish networks with NGOs and governmental agencies to amplify the impact of sustainability initiatives. Managers should also explore the opportunities for adaptation of climate friendly technologies and processes. Sixth, government agencies should take initiatives to foster a mindset among SMEs owners by emphasizing how sustainability aligns with long term profitability. Finally, government agencies should develop policies to monitor and evaluate environmental initiatives and performance of SMEs, which ensures improvement and adaptation to environmental initiatives. These research implications contribute to long term success and sustainable development of SMEs.

Reviewer 2 Comment Author's Response

1 Thank you for the opportunity to review this very interesting piece. While I believe your work is worth publishing, there are several issues that I would like to address prior to this. 

Thank you for your comments. We have improved the article following your instructions.

2 First Extensive editing of the English language and style is required. 

Thank you for your comment. We have proofread our article.

3 The title is confusing. Please re-write. 

Thank you for your comment. We have changed the title as follows:

Barriers to adaptation of environmental sustainability in SMEs: A qualitative study

4 Abstract: Please concise abstract of the study. 

Thank you for your comment. We have improved the abstract of the study, as given below:

Abstract 

This study examines the antecedents of environmental sustainability in small and medium enterprises (SMEs) of a developing country and explores the specific internal and external factors for environmental sustainability. The study focused on SMEs in Balochistan, Pakistan, utilizing convenience and purposive sampling techniques to select a sample size of 30 SMEs. In-depth qualitative interviews were conducted using a semi-structured questionnaire. The results of the study revealed that lack of finance and education are major barriers to recognizing and addressing environmental sustainability issues, along with the lack of government support and regulations to ensure compliance with environmental safety laws, hence leading to low concern for sustainability practices among SMEs. Awareness and attitude of SME owners/managers, along with customer demand and government policies, influence the adoption of environmental sustainability practices. Overcoming financial constraints and promoting cooperation among stakeholders are key to fostering sustainable practices in SMEs. This research makes an important contribution to the sustainable management literature by providing new and in-depth insights into the barriers that impede environmental sustainability in SMEs of developing countries.

5 Introduction 

- Set the stage for context, and history, and summarize what you’re going to do. Leave the reader with some impression as to the importance of the topic.

- There are many writing errors such as “Even though the research on environmental sustainability is growing, there is gap in how SMEs can achieve environmental sustainability especially in context of recourse constraint developing country.” What is this? Where is in/of?

- Why there is a need to explore the barriers and limitations of a developing country? There should be a strong background please pay attention to this part. 

Thank you for your comments. We have improved the introduction by following your suggestions. 

Page 1, Lines 30-36

SMEs have not received significant attention in the ongoing global discussion about sustainable development [4, 5]. It is evident that large organizations impact environment. It is also crucial to recognize the role of SMEs as well. SMEs impact on environment become clearer when we consider their collective impacts. Globally, over 95% of businesses are SMEs [6]. Although each SME may have little impact, the cumulative effect of SMEs can be substantial due to their large number [4]. It is often mentioned that SMEs are responsible for major pollution [7].

Page 1, Lines 40-45; Page 2, Lines 1-7

Researchers have been supporting the adaptation of sustainability practices [11, 3]. However, literature has not adequately addressed the sustainability issues faced by SMEs in developing countries as it has with developed ones [12]. This situation has been widely acknowledged in the literature [4, 5]. Wang et al. [10] shed light on the importance of sustainability initiatives in developing countries and the unique barriers and motivational factors. Similarly, Jabbour et al., [4] emphasized the significance of sustainability in developing countries and the need to study strategies for achieving sustainable practices. Additionally, Purwandani & Michaud [13] recognized the necessity to delve into sustainability matters within SMEs as it impacts business performance and presents new market opportunities.

Page 2, Lines 14-27

In this study we address the gap by exploring the internal and external factors that contribute to achieve the environmental sustainability in SMEs of Pakistan. The present study recognizes the existence of environmental alerts in Pakistan, particularly in Balochistan that comprise great environmental concern and weak governance [15, 16, 17]. Hence, this research significantly contributes to the discipline of sustainability. Further, the research framework of this study encompasses environmental orientation, responsible environmental management, and eco-friendly practices, which are grounded on strategic competencies such as pollution prevention, product stewardship, and sustainable development of natural resource-based view (NRBV) theory [18, 19]. Therefore, this study provides comprehensive understanding of internal and external dimensions of SMEs that need to be nurtured to improve environmental sustainability. While extending the previous research Baah et al. [18], this research provides new insight for top management, policymakers, governmental and non-governmental organizations (NGOs) for achieving environmental performance in SMEs of developing countries.

6 Literature review

- You used too many headings. Please read some articles to get a paper writing idea.

- The literature review in the study should be like a logical story that tells the readers about the gap that you intend to fill. What does the existing literature tell us? What do we know and don't know based on literature? 

Thank you for your comment. We have rewritten the literature review and incorporated your instructions on page 2, lines 44-47; page 3, lines 1-50; page 4, lines 1-21; page 5, lines 1-45; page 6, lines 1-27.

Page 2, Lines 44-47; Page 3, Lines 1-50; Page 4, Lines 1-21; Page 5, Lines 1-45; Page 6, Lines 1-27

SMEs prioritize survival and cost reduction that may not align with environmental sustainability [25]. The relationship between environmental sustainability and economic outcomes is complex and influenced by multiple factors. However, engaging in environmentally efficient practices can lead to long-term benefits and increased revenue through recycling, reuse and reduce [26]. Hence, it raised question whether eco-efficiency could increase profits for SMEs [27]. Researchers suggest that adaptation of sustainable practices and ecological partnerships can lead to financial success for SMEs [28]. Moreover, Hossain et al. [29] conducted a systematic literature review of articles published during 2009–2020 and identified 87 drivers of environmental sustainability and categorized them under internal and external dimensions. Researchers suggested that SMEs should focus on internal and external factors for implementation of sustainable practices. Hence, researchers identified internal and external barriers suitable to the context of this study. Table 1 summarizes the barriers and opportunities for SMEs. 

In reality, adopting sustainability requires resources, however, SMEs already facing several challenges [30]. On the other side, SMEs strive to gain flexibility and adaptabil

---

## [Editor Report · Decision Letter 1]

29 Jan 2024

Barriers to Adaptation of Environmental Sustainability in SMEs: A Qualitative Study

PONE-D-23-22784R1

Dear Dr. Khan,

We’re pleased to inform you that your manuscript has been judged scientifically suitable for publication and will be formally accepted for publication once it meets all outstanding technical requirements.

Kind regards,

Agnieszka Konys, Ph.D.

Academic Editor

PLOS ONE
---

## [Editor Report · Acceptance letter]

29 Apr 2024

PONE-D-23-22784R1 

PLOS ONE

Dear Dr. Khan, 

I'm pleased to inform you that your manuscript has been deemed suitable for publication in PLOS ONE. Congratulations! Your manuscript is now being handed over to our production team.

Kind regards, 

on behalf of

Dr. Agnieszka Konys 

Academic Editor

PLOS ONE